# Extracellular Vesicles Derived from MDA-MB-231 Cells Trigger Neutrophils to a Pro-Tumor Profile

**DOI:** 10.3390/cells11121875

**Published:** 2022-06-09

**Authors:** Carolinne Amorim, Clara Luisa Docasar, Daniel Guimarães-Bastos, Ana Clara Frony, Christina Barja-Fidalgo, Mariana Renovato-Martins, João Alfredo Moraes

**Affiliations:** 1Laboratório de Biologia RedOx, Universidade Federal do Rio de Janeiro, Rio de Janeiro 60440-593, Brazil; carolinne28amorim@hotmail.com (C.A.); clara.docasar@gmail.com (C.L.D.); daniel_gbastos@yahoo.com (D.G.-B.); 2Laboratório de Farmacologia Celular e Molecular, Universidade do Estado do Rio de Janeiro, Rio de Janeiro 23968-000, Brazil; clarafrony@gmail.com (A.C.F.); barja-fidalgo@uerj.br (C.B.-F.); 3Laborotário de Imunologia e Metabolismo, Universidade Federal Fluminense, Niterói 24220-900, Brazil; marianarenovato@id.uff.br; 4Instituto de Ciências Biomédicas, Universidade Federal do Rio de Janeiro, Rio de Janeiro 60440-593, Brazil

**Keywords:** extracellular vesicles, breast cancer, inflammation, tumor-associated neutrophils

## Abstract

Immune system cells, including neutrophils, are recruited by the tumor microenvironment as a site of chronic inflammation and begin to favor tumor growth. Neutrophils present in the tumor site are called tumor-associated neutrophils (TAN) and can present two phenotypes: N1 (antitumor) or N2 (pro-tumor). Evidence shows the high capacity of immune system cells to interact with extracellular vesicles (Evs) released by tumor cells. Evs can modulate the phenotype of cells within the immune system, contributing to tumor development. Here, we investigated the role of MDA-MB-231-derived Evs upon the polarization of neutrophils towards an N2 phenotype and the underlying mechanisms. We observed that neutrophils treated with Evs released by MDA cells (MDA-Evs) had their half-life increased, increased their chemotactic capacity, and released higher levels of NETs and ROS than neutrophils treated with non-tumoral Evs. We also observed that neutrophils treated with MDA-Evs released increased IL-8, VEGF, MMP9, and increased expression of CD184, an N2-neutrophil marker. Finally, neutrophils treated with MDA-Evs increased tumor cell viability. Our results show that MDA-Evs induce an N2-like phenotype, and the blockage of phosphatidylserine by annexin-V may be an essential agent counter-regulating this effect.

## 1. Introduction

Neutrophils are the most abundant leukocytes within the immune system, comprising 50 to 70% of total blood cells, being the first to migrate to the site of inflammation [1,2]. Chronic inflammation is a central component of tumor development and progression [3]. Regarding the tumor microenvironment cells, tumor-associated neutrophils (TAN) gain attention because of their antagonistic profiles [4,5,6,7,8,9,10,11]. These TAN present a duality of anti-tumor (TAN-N1) or pro-tumor (TAN-N2) phenotypes. TAN-N1 is more cytotoxic to tumor cells, releasing increased amounts of pro-inflammatory factors, such as CD195 (FAS), TNF-α, ICAM-1, and CCL3. At the same time, TAN-N2 presents an anti-inflammatory profile, exhibiting higher levels of CD184 (CXCR4), arginase-1, CCL2, CCL5, VEGF, IL-8, and MMP-9, supporting tumor growth, invasion, and metastasis [9].

Initially, reactive oxygen species (ROS) production from neutrophils was related to the phagocytosis process to defend against pathogen invasions. On the one hand, the release of ROS by neutrophils is essential for the death of pathogens; on the other hand, it can also be fundamental for determining cellular mechanisms favored by inflammation, such as cell proliferation and carcinogenesis [12,13].

Extracellular vesicles (Evs) are crucial elements sustaining the communication among biologically active cells through the exchange of nucleic acids (circRNA, mRNA, miRNA), lipids, and proteins, acting on intracellular signaling [14,15]. Cells exacerbate their EV production under oxidative stress conditions, such as cancer [16]. Externalization of phosphatidylserine in Evs’ lipid membrane is one of the main mechanisms of their release [14,17]. Previously, our group has demonstrated that melanoma-derived Evs induce neutrophils to a pro-tumoral phenotype [18]. Here, we have investigated whether Evs released by breast cancer cells (MDA-MB-231) polarize human neutrophils towards a TAN-N2 pro-tumoral phenotype, and if a phosphatidylserine ligand, annexin-v, could prevent this effect.

## 2. Materials and Methods

### 2.1. Cell Culture and EV Isolation

Human non-tumor cell line MCF10 and human breast carcinoma cell line MDA-MB-231 were obtained from ATCC (Manassas, VA, USA). The cells were grown in DMEM/F12 supplemented with 23 mM NaHCO3, 21.8 mM HEPES, 60 mg/L penicillin, 100 mg/L streptomycin, and 10% FBS, pH 7.2, at 37 °C in a humidified atmosphere with 5% CO_2_ under mycoplasma-free conditions. The Evs contained in the serum were removed using the same EV isolation protocol (described below). The Evs were discarded, and the supernatant was collected to be used in cell culture. When the cells reached 100% confluence (7 × 10^6^), the medium containing 10% FBS was changed to 1%, and the cells were kept for 24 h under the same conditions. Conditioned mediums (CM) from tumor and non-tumor cells were collected after 24 h and centrifuged (1000× *g* at 10 min) to remove cell debris. Afterward, the CM-containing Evs were ultracentrifuged at 4 °C for four hours at 100,000× *g*, and the supernatant was discarded. The vesicles were resuspended in the original volume in Hank’s Balanced Salt Solution (HBSS) for ROS assay, incomplete culture medium RPMI-1640 for other assays, a buffer of radioimmunoprecipitation lysis (RIPA) for western blotting of Evs, or binding buffer to annexin-V for characterization of Evs in flow cytometry. After isolation, the Evs were stored for up to 6 months at −80 °C.

### 2.2. Isolation of Human Neutrophils

Neutrophils were collected from healthy donors (approval #38257914.7.0000.5259 from the Pedro Ernesto Hospital Ethics Committee, UERJ, RJ, Brazil) in BD vacutainer tubes with EDTA, isolated in a continuous gradient of ficoll (1:2) and centrifuged at 750× g for 40 min. Plasma and mononuclear cells were discarded, and 6% dextran in sterile PBS was added for red blood cells (RBCs), decantation at 37 °C, 5% CO_2_, for 40 min. The neutrophils with RBCs were collected and centrifugated at 1000× *g* for 10 min, and the RBCs were lysed in saline buffer (0.2% and 1.6% NaCl, respectively). Neutrophils were isolated and counted in a Neubauer camera using trypan blue. Cells were resuspended in incomplete RPMI-1640, HBSS, RIPA lysis buffer, or PBS, depending on the experiment.

### 2.3. In Vitro TAN-Like Polarization

Neutrophils were placed in conical-bottom tubes for 3 h at 37 °C and under a 5% CO_2_ atmosphere as previously described [18]. The experimental groups used were: untreated (incomplete RPMI-1640 only), annexin-V (10 nM), LPS (10 µg/mL), MCF10 Evs (30% *v/v*), MDA-MB-231 Evs (30% *v/v*) with or without pretreatment with annexin-V for 15 min.

### 2.4. Cell Migration Assay

Cell migration assay was performed using a 48-well Boyden chamber (Neuroprobe Inc., Gaithersburg, MD, USA) with polycarbonate membranes in 5 µm pores, as previously described [19]. Isolated neutrophils (10^6^ cells/mL) were added to the upper wells, and the stimuli were added to the lower wells, as previously described in the previous section. Migration towards the medium solely (RPMI-1640) was used as a negative control for the experiment, while migration towards fMLP (100 nM) was used as a positive control. After 60 min of migration at 37 °C and under a 5% CO_2_ atmosphere, the membranes were removed, fixed, and stained with a Diff-Quick™ kit (Panotico-Laborclin, Pinhais, PR, Brazil). The number of neutrophils that migrated to the underside of the membrane was counted in at least five random fields (1000× magnification) under light microscopy (Olympus BX41, Tokyo, Japan). Results are representative of at least three independent experiments performed in quintuplicates and are expressed as the mean ± SEM number of neutrophils per field.

### 2.5. Whole Cell Extraction Neutrophils

Neutrophils were incubated as previously described. After polarization, the tubes were centrifuged at 1000× g for 10 min, the supernatant discarded, and the cells resuspended in RIPA lysis buffer containing SIGMAFAST™ Protease Inhibitor (cocktail tablet, without EDTA, Sigma Aldrich, St. Louis, MO, USA), DNAse (2 mg/mL) and orthovanadate (1 mM) on ice for 20 min.

### 2.6. Whole Cell Extraction MDA-MB-231

In a 24-well plate, 10^5^ cells/well were plated in DMEM/F12 10% FBS. Subsequently, neutrophils were purified, and the groups were prepared as previously described. After polarization, neutrophils were centrifuged at 1000× *g* for 10 min, the supernatant discarded, and the cells resuspended in RPMI-1640 10% FBS. The culture medium was removed from the plate wells containing MDA-MB-231, and a 0.4 µm transwell was allocated to all wells. Neutrophils (10^6^ cells/insert) were placed on top of the inserts, and after 24 h of stimulation, the inserts with neutrophils were discarded, as well as the medium remaining in the plate. The tumor cells were lysed with RIPA lysis buffer containing SIGMAFAST™ Protease Inhibitor, DNAse (2 mg/mL), and orthovanadate (1 mM) on ice for 20 min.

After lysis, proteins were measured by the BCA Protein Assay Kit (Thermo Fisher Scientific, Waltham, MA, USA), and a 20% (*v/v*) sample buffer was added for blotting (33.3% glycerol, 16.7% β-mercaptoethanol, 10% SDS, 0.33% 10 M NaOH and bromophenol blue). Samples were boiled at 100 °C for 5 min and stored at −20 °C until use.

### 2.7. Electrophoresis and Western Blotting Assay

Cell lysates (10–20 μg of protein) were fractionated by polyacrylamide gel electrophoresis 10–12% (SDS-PAGE) denaturant, and separated according to different molecular weights (Standard Precision Plus Protein Kaleidoscopic™, Bio-Rad, Hercules, CA, USA). The proteins were transferred (25 V, 1.0 A, and 30 min) using a transfer buffer with methanol and polyvinylidene difluoride membrane (Immun-Blot^®^ PVDF, Bio-Rad) on Trans-Blot Turbo™ equipment (Bio-Rad). Then, the membranes were washed with 5% BSA (Sigma-Aldrich, St. Louis, MI, USA) in 0.1% T-TBS and incubated overnight at 4 °C in the presence of the following primary antibodies: anti-arginase (gt5811-rabbit-1:1000) (Thermo Fisher Scientific, Waltham, MA, USA), anti-caspase-3 (sc7148-rabbit-1: 250) (Santa Cruz Biotechnology, Dallas, TX, USA), or anti-β-actin (a5441-mouse-1:5000) (Sigma-Aldrich). The membranes were incubated for at least one hour with a specific peroxidase-conjugated anti-rabbit secondary antibody (ab6789-mouse or ab6721-rabbit-1:10,000) (Abcam, Cambridge, MA, USA). Immunoreactive proteins were visualized by detecting their chemiluminescence through ECL (GE) solution. Membranes were developed and photographed using ImageQuant™ LAS500 (GE Healthcare, Buckinghamshire, England), and densitometry was quantified using ImageJ software.

### 2.8. Apoptosis Assay

Neutrophils were treated for 20 h, as previously described. After polarization, the tubes were centrifuged at 1000× *g* for 10 min, the supernatant discarded, and the cells resuspended in PBS. Apoptosis was evaluated in two different ways. (A) Morphology: 2 × 10^4^ cells were centrifuged in CytoSpin to mount slides at 500× *g* for 5 min, fixed, and stained with Diff-Quick™ staining. The number of neutrophils that underwent apoptosis was evaluated through the pyknotic nucleus and were counted in at least three random fields (1000× *g* magnification) under light microscopy (Olympus BX41, Tokyo, Japan). The results are representative of at least three independent experiments performed and are expressed as mean ± SEM of the number of neutrophils per field. (B) Flow cytometry: 2 × 10^6^ cells were evaluated for phosphatidylserine exposure, in which cells were incubated with binding buffer for annexin-V (10 mM Hepes, 140 nM NaCl, 2.5 mM CaCl_2_, and 0.75 mM MgCl_2_, pH 7.4) and labeled with annexin-V conjugated to fluorescein-isothiocyanate (FITC) (1:50) for 20 min at room temperature, in the absence of light. After incubation, propidium iodide (PI, 10 μg/mL in binding buffer) conjugated to PE (Sigma-Aldrich) was added to investigate necrosis and then analyzed by flow cytometry (Accuri C6™, BD Bioscience, San Jose, CA, USA) for the identification of cells with positive events for annexin V (annexin V+/PI™ and annexin V+/PI+ cells).

### 2.9. Reactive Oxygen Species Assay

Neutrophils were resuspended in HBSS (without phenol red) and placed in conical-bottom tubes containing a specific probe for ROS detection (CM-H2DCFDA or DAF–10 µM). The cells were incubated at 37 °C and 5% CO_2_ for one hour to internalize the probes. After this time, the tubes were centrifuged (1000× *g* at 10 min) to remove the probe not internalized in the cells. Neutrophils were treated as previously described, and immediately, in a black 96-well plate, stimulated cells (3 × 10^5^ cells/mL) were monitored for probe oxidation using an EnVision™ Multilabel Plate Reader at wavelengths of excitation and emission of 495 and 525 nm, respectively.

### 2.10. Neutrophil Extracellular Trap Assay (NETs)

Neutrophils (6 × 10^6^ cells/mL) were stimulated according to the groups described in the previous section and were incubated at 37 °C and 5% CO_2_ for three hours. After polarization, DNAse (2 µg/mL) was added to all groups and kept under the same conditions for 20 min. The supernatant was collected and kept on ice, and the DNA was quantified in the NanoDrop™.

### 2.11. Quantification of Cytokines

The amount of IL-8, VEGF, CCL5, and CCL2 present in the neutrophils supernatants was quantified by an ELISA Development Kit from Peprotech (Rocky Hill, NJ, USA) (IL-8 #900-TM31; VEGF #900-TM10; CCL2 #900-TM31 and CCL5 #900-M33), according to the manufacturer’s instructions. The neutrophils were polarized for three hours, and their supernatants containing the stimuli were removed; the cells were washed and maintained for more than two hours to release the cytokines.

### 2.12. Analysis of CXCR4 and Fas Markers

Neutrophils (1.5 × 10^6^ cells/mL) were incubated as described above for 3 h. Afterward, neutrophils were centrifuged (1000× *g* for 10 min), suspended in flow cytometry buffer (PBS solution containing 2% FBS, 5 mM EDTA), and incubated for 15 min with FITC-conjugated anti-CD95 (Fas), and anti-CD184 conjugated to allophycocyanin (APC) (CXCR4) (BioLegend, San Diego, CA, USA). The expression of these molecular markers was analyzed using an Accuri™ C6 Flow Cytometer.

### 2.13. Zymography for MMP-9

Neutrophils (6 × 10^6^ cells/mL) were incubated as described above for three hours. After polarization, tubes were centrifuged at 1000× *g* for 10 min, and the supernatant was stored. Then, the supernatant was subjected to centrifugation in a SpeedVac (Savant SPD111; Thermo Fisher Scientific) for 4 h to concentrate the proteins. The neutrophil supernatants were resuspended in a Triton lysis buffer (50 mM HEPES buffer, pH 7.5 containing 250 mM NaCl, 1% Triton X-100, and 10% glycerin) supplemented with protease inhibitors.

Then, the samples were subjected to SDS-PAGE, as previously described [18]. Briefly, the gels (SDS-PAGE, 7.5%) were copolymerized with 2 mg/mL gelatin. The gels were further washed in renaturation buffer (2.5% Triton X-100 in 50 mM Tris–HCl, pH 7.5) and incubated at 37 °C for 24 h in substrate buffer (10 mM Tris–HCl buffer, pH 7.5; 5 mM CaCl_2_; and 1 mM ZnCl_2_). The gels were then stained with a 30% methanol/10% acetic acid solution containing 0.5% Coomassie blue, and discolored using the same solution without dye. Areas of enzymatic activity appeared as clear bands over the dark background. The image was digitally inverted so that the integration of the bands was reported as positive values. Matrix metallopeptidase (MMP) production was measured by scoring the intensity of bands and arbitrary OD units via image analysis performed with ImageJ software.

### 2.14. MTT Assay

MDA-MB-231 cells were plated in a 96-well plate with DMEM/F12 10% FBS. When the cells reached 90% confluence, the neutrophil isolation/purification process was started (2.5 × 10^6^ cells/ mL). Neutrophils were stimulated according to the groups already described in the previous section. After polarization, the tubes were centrifuged at 1000× *g* for 10 min, the supernatant discarded, and the cells resuspended in the same volume in RPMI-1640 10% FBS culture medium. The 96-well plate containing MDA-MB-231 had the 10% DMEM medium discarded. Each group of neutrophils was plated on the wells of the plate-containing tumor cells, respecting the ratio of one tumor cell for every ten neutrophils. The co-cultures of MDA-MB-231 with neutrophils were incubated at 37 °C and 5% CO_2_ for 24 h. We also incubated only neutrophils with MTT to exclude the possibility of neutrophils metabolizing MTT. In the last four hours of co-culture, MTT solution (5 mg/mL) in PBS was added, and the plate remained under the same conditions. MTT metabolization by tumor cells was monitored by the formation of insoluble salts, dissolved with isopropyl alcohol, and absorbance read at 570 nm.

### 2.15. Mitochondrial Membrane Potential Assay

In a 24-well plate, 5 × 10^4^ cells/well were plated in DMEM/F12 10% FBS. In the next step, the neutrophil purification was started, and the groups were prepared as previously described. After polarization, neutrophils were centrifuged at 1000× *g* for 10 min, the supernatant discarded, and the cells resuspended in RPMI-1640 10% FBS. The culture medium was removed from all wells of the plate containing MDA-MB-231, and a 0.4 µm transwell was allocated to all wells. Neutrophils (5 × 10^5^ cells/insert) were placed on top of the inserts. After 20 h of stimulation, the inserts with neutrophils were discarded, as well as the medium remaining in the plate, and the tumor cells’ mitochondrial transmembrane potential was assessed in EnVision™ using JC-1 probe (10 μg/mL, Thermo Fisher Scientific), according to the manufacturer’s instructions.

### 2.16. Statistical Analysis

Data are expressed as means ± standard error of the mean. The comparisons of means to quantitative variables between groups were performed using a Student’s *t*-test or ANOVA one-way test, followed by a Bonferroni post-test. A value of *p* < 0.05 was considered statistically significant. Statistical evaluation was performed using GraphPad Prism software Version 7.0 (GraphPad Software, San Diego, CA, USA).

## 3. Results

### 3.1. Effect of MDA-EVs on Human Neutrophils

EVs derived from breast tumor cells (MDA-MB-231) and non-tumor cells (MCF10) were isolated. The nanometer-scale size distribution was determined using dynamic light scattering, where 86.2% of MCF10-EVs have a size of 183.2 nm, and 96.3% of MDA-EVs have a size of 368.4 nm (Appendix A). As already described in the literature [14], stress conditions, such as cancer, significantly increase the release of EVs, mainly exosomes. The exosomes were characterized for the presence of exosomal markers CD63 and syntenin-1 by Western blotting (Appendix A). As in the previous Zetasizer analysis, it was possible to observe a more significant presence of exosomes from the CM of tumor cells (MDA-MB-231). To quantify larger EVs (microparticles), we performed flow cytometry analysis. Events gated under 2 µm beads area and annexin-V positive were considered EVs (microparticles). Our results show an increase in 30% of events from MDA-MB-231-EVS compared to non-tumor EVs, or 600 EV/µL from MDA-EVs and 470 EV/µL from MCF10-EVs (Appendix A). 

After 20 h of culture, tumor EVs increased neutrophil viability, as we observed through the morphological analysis of pyknotic nuclei (Figure 1A), and the same result was observed when we performed flow cytometry analysis (Figure 1B). When neutrophils were treated with MCF10-EVs or MDA-EVs pre-treated with annexin-V, neutrophil viability was not enhanced, suggesting that MDA-EVs protect neutrophils from spontaneous apoptosis via phosphatidylserine.

Then, neutrophils were challenged to migrate for one hour (chemotaxis) in the direction of tumor and non-tumor EVs, and we observed that MDA-EVs favor neutrophil migration. The fMLP was used as a positive control for the assay, demonstrating higher cell migration concerning this stimulus than the others (Figure 1C). The chemotactic effect was not observed when neutrophils were induced to migrate against MCF10-EVs and MDA-EVs blocked by annexin-V. We also evaluated neutrophil extracellular DNA release when the cells were treated with EVs. After three hours, we observed that neutrophils treated with LPS and MDA-EVs released extracellular DNA. This increase suggests a higher production of neutrophil extracellular traps (NETs) in neutrophils treated with LPS and MDA-EVs. Extracellular DNA release was not observed when neutrophils were treated with MCF10-EVs or MDA-EVs pre-treated with annexin-V, suggesting the importance of phosphatidylserine in tumor EVs (Figure 1D). These data confirm the activation of neutrophils by MDA-EVs through protection from spontaneous apoptosis, chemotaxis induction, and increased release of extracellular DNA.

### 3.2. Effect of MDA-EVs on Human Neutrophils Redox State

Neutrophils release ROS during inflammatory processes, supporting tumor growth [9,12]. According to this idea, our results demonstrate that after one hour, neutrophils stimulated with MDA-EVs show an increase in ROS production compared to neutrophils stimulated with MCF10-EVs. ROS production was not observed when neutrophils were exposed to MDA-EVs pre-treated with annexin-V (Figure 2A). Conversely, neutrophils treated with tumoral EVs released lower nitric oxide (NO) levels than those treated with non-tumoral EVs (Figure 2B). Thus, we observed that tumoral EVs could activate neutrophils by increasing ROS production; however, we also have observed reduced NO production.

### 3.3. In Vitro Induction of Neutrophil N2-Like Phenotype

Several molecular markers such as IL-8, IL-6, VEGF, CCL2, CCL5, STAT-3, arginase-1, MMP-2, MMP-9, and CXCR4, among others, are characteristic of TAN N2. We observed an increase in IL-8 (Figure 3A) and VEGF (Figure 3B) in the CM of neutrophils stimulated with MDA-EVs for three hours, which was prevented by the pre-treatment of tumoral EVs with annexin-V (Figure 3A,B). Conversely, the levels of CCL2 and CCL5 released by neutrophils were not affected by the treatment with tumoral EVs (Figure 3C,D). We also observed increased arginase-1 expression in neutrophils treated with MDA-EVs compared to those treated with MCF10-EVs, and this increase was not observed when MDA-EVs were pre-treated with annexin-V (Figure 3E). Furthermore, we noticed that the treatment with tumoral EVs increased bioactive MMP-9 release by neutrophils compared to those treated with non-tumoral-EVs, which was abolished when MDA-EVs were pre-treated with annexin-V (Figure 3F). To confirm the induction of the N2-like profile by MDA-EVs, we investigated the expression of specific surface markers of TAN N1 (CD95-Fas receptor) or TAN N2 (CD184-CXCR4 receptor). Our results reveal the same levels of CD95 expression between the groups analyzed (Figure 3G). However, the treatment of neutrophils with MDA-EVs increased CD184 levels, which was not observed by neutrophils treated with MCF10-EVs and MDA-EVs pre-treated with annexin-V (Figure 3H). These results together demonstrate that MDA-EVs can induce the N2-like or pro-tumor profile of neutrophils, and that phosphatidylserine’s blockage can abrogate the effect of MDA-EVs.

### 3.4. Effect of N2-Like Neutrophils on Breast Tumor Cells (MDA-MB-231)

We investigated the role of N2-like neutrophils upon breast tumor cell viability. For this, cells were treated with neutrophils previously incubated or not with EVs. Our results demonstrate that N2-like neutrophils increase breast tumor cell viability (MDA-MB-231), which was not observed when neutrophils were stimulated with MCF10-EVs or MDA-EVs pre-treated with annexin-V (Figure 4A). We also evaluated neutrophils alone; however, we observed that they could not metabolize MTT. Concurrently, our results demonstrate a high expression of pro-caspase by tumor cells treated with N2-like neutrophils, suggesting lower apoptosis of breast tumor cells (Figure 4B). We also investigated mitochondrial membrane potential through the JC-1 probe to confirm these findings. Thus, we observed that N2-like neutrophils inhibit MDA-MB-231 mitochondrial membrane potential enhancement induced by neutrophils. We also observed that neutrophils treated with LPS strongly disrupted MDA-MB-231 mitochondrial membrane potential (Figure 4C). Together, these results show that neutrophils with a pro-tumoral profile can increase the viability of breast tumor cells.

## 4. Discussion

One of the main characteristics of the carcinogenic process is the recruitment of immune system cells, triggering chronic inflammation, which has recently been established as one of the hallmarks of cancer [3]. However, in this context, the role of tumor-associated neutrophils (TAN) is not fully understood. Under inflammatory conditions, neutrophils, which represent the most significant percentage of circulating leukocytes, are the first cells to migrate to the injured site to initiate tissue repair. Since 1863, neutrophils have been described within the tumor microenvironment [20]; several pieces of evidence have continuously demonstrated its relationship with a poor prognosis in some types of carcinomas, such as melanoma and hepatocellular [21]. However, in colorectal carcinoma, the presence of neutrophils is correlated with an improvement in patient survival [22]. Such duality can be explained by the neutrophils’ capacity to acquire different molecular signatures, depending on the microenvironment [8,12,23,24,25].

The capacity of tumor cells to release EVs as a form of intercellular communication is extensively described [14,15,16,17,26,27]. EVs can interact with immune system cells [28] and even activate neutrophils towards a pro-tumoral phenotype [18,29].

Some studies have already reported the effects of the interaction between EVs from tumor cells with cells of the immune system, such as the ability of EVs from breast cancer and melanoma cells to inhibit the proliferation of NK cells [30]. Furthermore, it was also demonstrated that EVs from ovarian cancer patients could induce the differentiation of T helper lymphocytes into regulatory T lymphocytes [31]. However, few data in the literature demonstrate the relationship of EVs with neutrophils, especially the relationship of EVs with the polarization of these immune system cells. There is also no description in the literature regarding the ability to block the effects of EVs on neutrophils. 

Recently, studies regarding phosphatidylserine have gained attention. For example, EVs from patients with ovarian cancer inhibit, in vitro, the activation of CD4+ and CD8+ T lymphocytes. The previous treatment of these EVs with an anti-phosphatidylserine antibody reversed this effect [32]. Additionally, in a murine model of melanoma, the blockage of EVs with annexin-V was shown to revert its effects upon TGF-b release by macrophages. [33].

In the present work, we investigated the effect of EVs released from MDA-MB-231 cells (metastatic breast cancer) on the activation of human neutrophils. Zhang et al. describe that EVs derived from gastric cancer cells induce neutrophils to a pro-tumor profile, increasing their migratory capacity [29]. Our results demonstrate that tumoral EVs protect neutrophils from spontaneous apoptosis, which remains activated within the tumor microenvironment, possibly contributing to tumor development. We also observed that MDA-EVs other than MCF10-EVs created a chemotactic environment, favoring the migration of neutrophils. 

The epithelial cell line MCF10 was used as a comparison control in this in vitro character study experiment. Due to the genetic alteration, they may present a pre-malignant character and do not reliably represent the healthy reality. However, this strain is the most used by the authors at experimental levels compared with breast tumor cells.

When classically activated by pro-inflammatory stimuli, neutrophils release NETs and ROS; corroborating this idea, we demonstrated that the treatment of neutrophils by MDA-EVs increased the release of NETs when compared to MCF10-EVs, as observed by Leal et al. [34]. 4T1-EVs from breast tumors also induce the formation of NETs. It is known that neutrophils secrete NETs to fight pathogens, with subsequent elimination of the agent causing the injury. However, during cancer, neutrophils secrete NETs, exacerbated due to the persistence of the injury, sustaining the development of tumor cells. The NET generation mechanisms and composition vary according to the inflammatory stimulus [35]. Neutrophils activated by a pro-inflammatory stimulus such as LPS produce elastase-enriched NETs.

In contrast, neutrophils induced by an anti-inflammatory stimulus such as EVs released by MV3—a melanoma cell—produce NETs poor in elastase [18]. Concurrently, the dual role of ROS released by neutrophils also depends on the origin of the reactive species produced [13]. Our results demonstrate high intracellular ROS production and low nitric oxide levels by neutrophils treated with MDA-EVs. We call attention to EVs, which can carry NOX subunits, inducing ROS formation and consequently modulating oxidative stress in target cells. Different types of stimuli modulate the redox balance of tumor cells. They release proteins and lipids within EVs to maintain their redox homeostasis, thus inducing oxidative stress in target cells [27]. When activated by MDA-EVs, the neutrophils released increased levels of IL-8 and VEGF, which is consistent with an N2-like phenotype [36]. Besides being an N2-like marker, IL-8 is a chemoattractant and activator of human neutrophils [37]. The increase of this cytokine by neutrophils may be sustaining the maintenance of high levels of neutrophils within the tumor microenvironment, which is consistent with chronic inflammation. Neovascularization further supports tumor growth, where VEGF plays a crucial role. VEGF is released by several immune system cells, including neutrophils [38].

Here, we demonstrated that the high levels of VEGF released by neutrophils are another crucial feature of the N2-like profile. We also assessed the ability of tumor-associated neutrophils to possibly recruit other immune system cells, and evaluated the release of CCL2/MCP-1 and CCL5/RANTES by neutrophils stimulated with MDA-EVs. However, we did not observe any difference between the groups (Figure 3A). CCL2 is a classical monocyte chemoattractant cytokine [39]. Its increase by neutrophils could indicate considerable signaling to monocytes in the tumor microenvironment that, once attracted, could be polarized to the M2-like phenotype with the aid of CCL5 [40], thus increasing the inflammatory response. We observed other important biomarkers already described in the literature [9]. Our results demonstrate high levels of arginase-1, which is consistent with the low production of NO by neutrophils treated with MDA-EVs, since arginase competes with nitric oxide for L-arginine [41]. Such combined effects suggest a decreased cytotoxicity by these N2-like neutrophils. Concurrently, the treatment of neutrophils with MDA-EVs increased the release of bioactive MMP9, supporting its pro-tumoral molecular profile. MMP9 is a metalloproteinase capable of cleaving several proteins present in the extracellular matrix to regulate matrix remodeling and, in the context of cancer, increase the proliferation, migration, and invasion of tumor cells to other tissues [42,43].

Therefore, an increased expression of MMP9 by neutrophils within the tumor microenvironment may increase tumor progression and development. Together, the increased levels of the surface marker CD184 (CXCR4) on neutrophils treated with MDA-EVs corroborate the polarization of those cells towards an N2-like phenotype. Our results demonstrate that, despite the treatment of VEs with annexin-V to block phosphatidylserine, CD95 did not increase in neutrophils—an N1-marker—rather, it prevented the polarization towards an N2-like phenotype. Consistently, the viability and levels of pro-caspase-3 in MDA-MB-231 cells were increased by N2-like neutrophils and prevented in the presence of annexin-V.

## 5. Conclusions

Together, our data suggest that MDA-EVs induce an N2-like phenotype in neutrophils marked by increased migration, viability, the release of NETs, ROS, extracellular DNA, IL-8, and VEGF, as well as increased arginase-1 expression, MMP9 activity, and mainly the ability to increase the viability of breast tumor cells (MDA-MB-231). The use of annexin-V may be a helpful strategy to minimize the effect of MDA-EVs on neutrophils through the blockage of phosphatidylserine (Figure 5).

## Figures and Tables

**Figure 1 cells-11-01875-f001:**
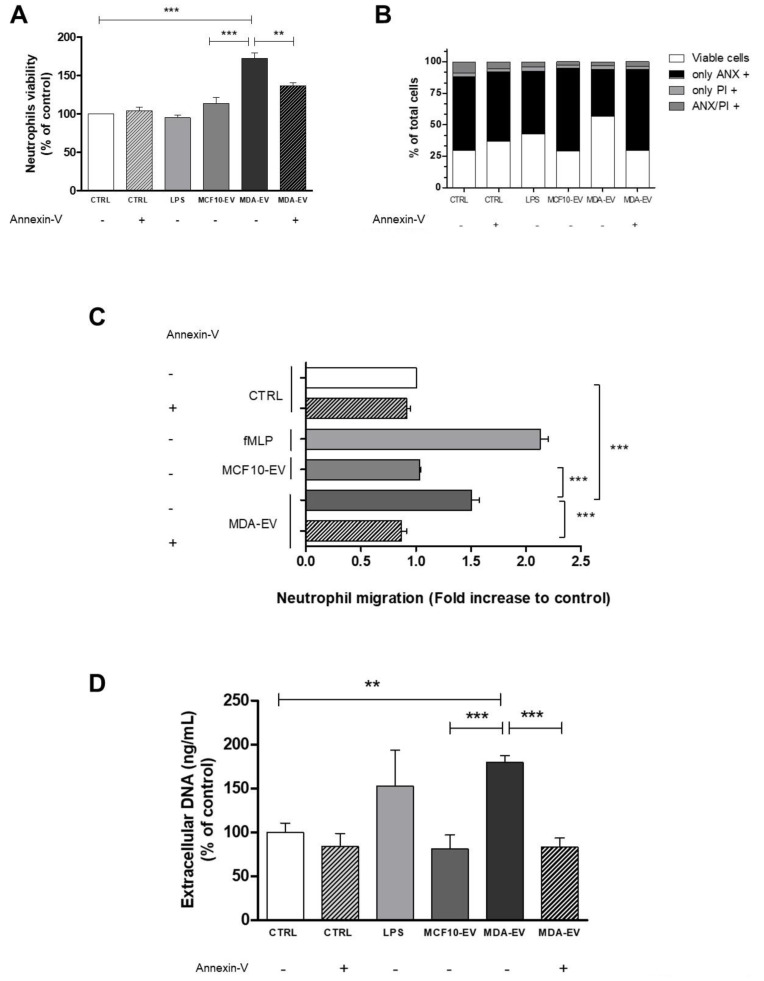
Effect of MDA-EVs on human neutrophils. (**A**) The nuclear morphology of neutrophils (magnification: 100×) and (**B**) phosphatidylserine membrane exposure was evaluated for apoptosis using annexin V-FITC and PI binding, respectively. Flow cytometry was used to identify apoptotic cells (annexin-V+/PI– and annexin-V+/PI+). (**C**) MDA-EVs induce neutrophil chemotaxis compared to MCF10-EVs. (**D**) The amount of extracellular DNA in the supernatant of neutrophils was quantified using NanoDrop^TM^. The annexin-V concentration was 10 nM. Data are expressed as mean + standard error (SEM). ** *p* < 0.005; *** *p* < 0.001. Results are representative of 3–6 independent experiments.

**Figure 2 cells-11-01875-f002:**
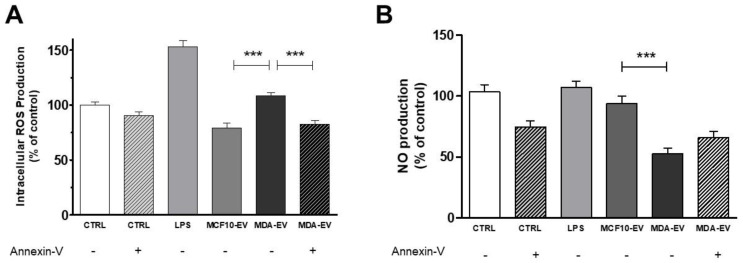
Neutrophils treated with MDA-EVs produce high levels of total ROS and low levels of Nitric Oxide. (**A**) Neutrophils were incubated for one hour with a CM-H2DCFDA probe, and intracellular total ROS production was monitored using a plate reader. (**B**) Neutrophils were incubated with a DAF-FMDA probe, and NO production was analyzed using a plate reader. The annexin-V concentration was 10 nM. Data are expressed as mean + standard error (SEM). *** *p* < 0.001. Results are representative of 8–22 independent experiments.

**Figure 3 cells-11-01875-f003:**
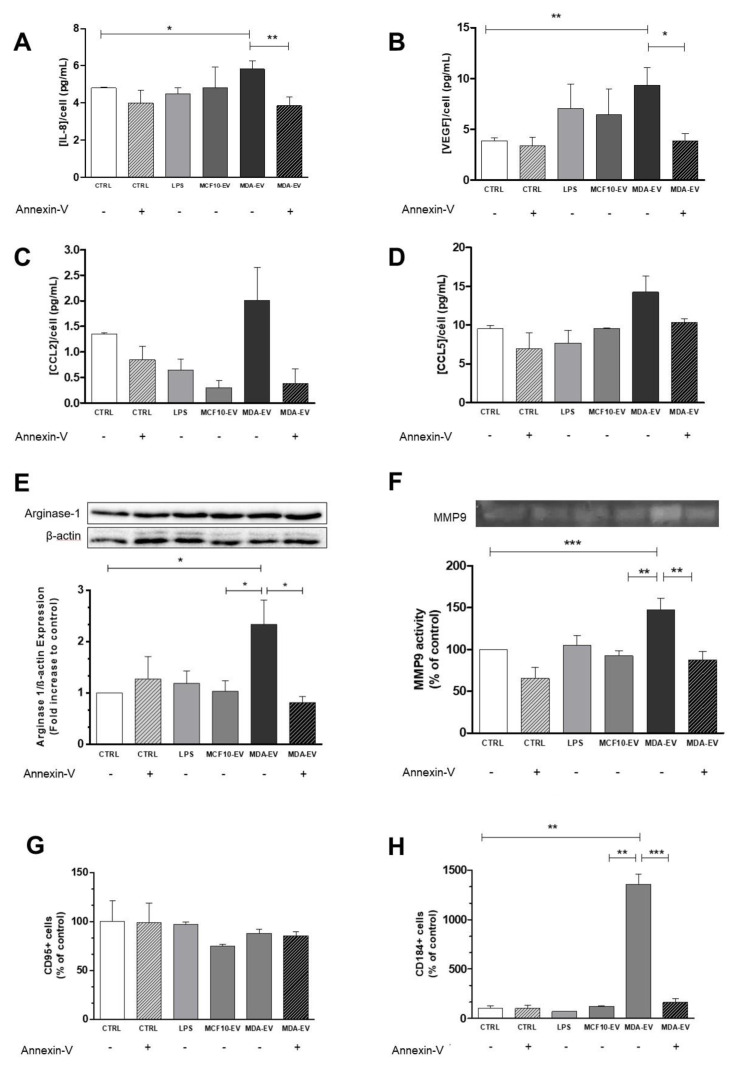
MDA-EVs induce neutrophils to N2-like phenotype. Neutrophils were polarized for three hours, washed, and conditioned media were collected after two hours. The release of (**A**) IL-8, (**B**) VEGF, (**C**) CCL2, and (**D**) CCL5 was evaluated by ELISA. (**E**) Neutrophils were lysed, and the protein extracts were separated using SDS-PAGE with 12% acrylamide gels to identify arginase-1 expression. In the representative image, contrast and brightness were adjusted to better visualize the bands. (**F**) The supernatant from polarized neutrophils was subjected to electrophoresis using SDS-page with 7.5% acrylamide and gelatin. The gels were then stained with a 30% methanol/10% acetic acid solution containing 0.5% Coomassie blue, and discolored using the same solution without dye. Areas of enzymatic activity appeared as clear bands over the dark background. (**G**) Neutrophils were incubated for 15 min with FITC-conjugated anti-CD95 (**H**) and APC-conjugated anti-CD184, following which these protein contents were assessed using flow cytometry. The annexin-V concentration was 10 nM. Data are expressed as mean + standard error (SEM). * *p* ≤ 0.05; ** *p* < 0.005; *** *p* < 0.001. Results are representative of 4–15 independent experiments.

**Figure 4 cells-11-01875-f004:**
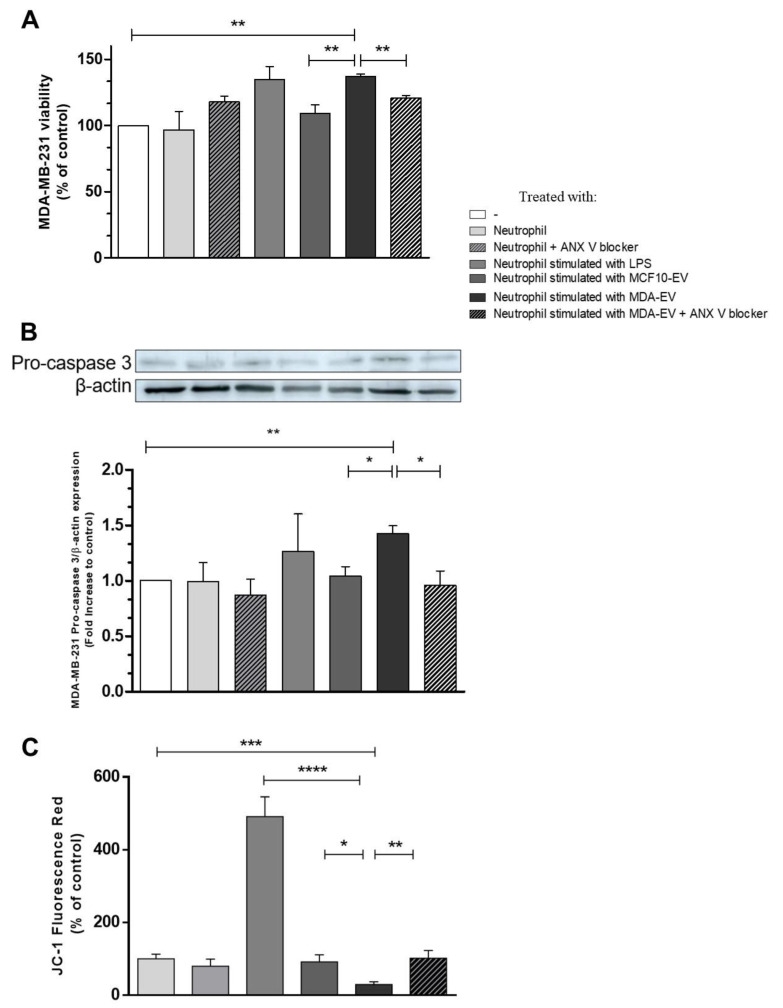
N2-like neutrophils increase tumor cell viability. (**A**) Polarized neutrophils were co-cultured with previously seeded MDA-MB-231 cells (1:10 ratio of MDA: neutrophils) for 24 h. Cell suspensions were incubated with MTT (5 mg/mL) during the last four hours, and post this, MTT metabolization by viable MDA cells was monitored using a plate reader at 570 nm. (**B**) MDA-MB-231 cells were primed by N1 or N2-like neutrophils, using a transwell insert. After 24 h, the tumor cells were lysed, and the protein extracts were separated using SDS-PAGE with 12% acrylamide gels to identify pro-caspase expression. The annexin-V concentration was 10 nM. Data are expressed as mean + standard error (SEM). (**C**) MDA-MB-231 cells were primed by N1 or N2-like neutrophils, using a transwell insert. After 24 h, the mitochondrial membrane potential was monitored using the JC-1 probe. * *p* ≤ 0.05; ** *p* < 0.005; *** *p* ≤ 0.001; **** *p* ≤ 0.0001 Results are representative of 3–5 independent experiments.

**Figure 5 cells-11-01875-f005:**
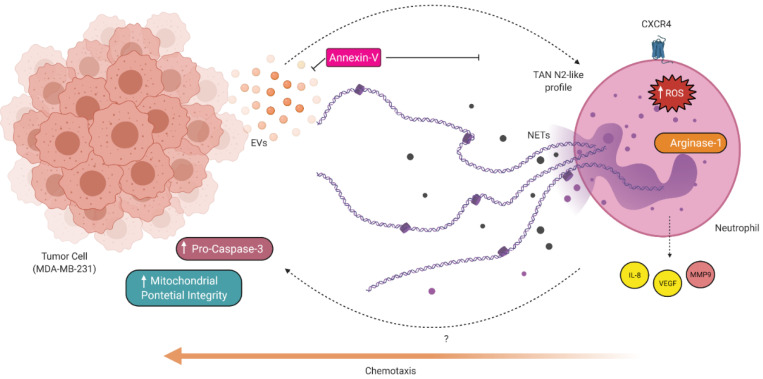
Proposed model for N2-like neutrophil polarization by EVs derived from breast tumor cells (MDA-MB-231). Tumor cells produce and release EVs, which create a chemoattractant environment for neutrophils, protect neutrophils from spontaneous apoptosis, induce the release of NETs and production of total intracellular ROS. Furthermore, neutrophils are induced to a pro-tumor N2-like phenotype, exhibiting increased molecular markers such as IL-8, VEGF, MMP-9, arginase-1, and CXCR4. Neutrophils can also increase the viability of breast tumor cells in vitro. Created with Biorender.com accessed on 6 January 2022.

## Data Availability

Not applicable.

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
