# Peer review of "Extracellular Vesicles Derived from MDA-MB-231 Cells Trigger Neutrophils to a Pro-Tumor Profile"

_cells, 2022, doi:10.3390/cells11121875_

Round 1

Reviewer 1 Report

The authors adressed my main concerns satisfactorily and I don't have more comments. 

Author Response

We thank the reviewer for the opportunity to improve the manuscript.

Reviewer 2 Report

The manuscript by Amorim and colleagues investigates the effect of extracellular vesicles (EVs) from breast cancer cells (MDA-MB-231) on neutrophils and how this interaction impacts the pathogenesis of breast cancer. The data points to a role of EVs stimulating the polarization of neutrophils towards the N2 type, which results in the stimulation of cancer cell proliferation. A role of phosphatidylserine on the EV surface is suggested. The study concept is very interesting and innovative and in general the conclusions are supported by the experimental evidence. The following aspects have to be addressed to strengthen the message of the article and improve the readability and clarity:

1- Section 2.1. Specify that MCF10 cells are non-tumorigenic.

2- Section 2.1. The method for EV isolation requires further explanation. How many cells were plated in order to obtained the CM? In which volume were the EVs resuspended? Please, specify which EV diluent was used for which application.

3- Section 2.1. For the generation of CM, cells were cultured in the presence of 1% FBS. FBS is a common source of EVs. Was the FBS previously treated in order to eliminate this potential interference?

4- The use of the expression “Annexin-V blocker” throughout the manuscript is confusing. Was actually “Annexin V” used to bind and, therefore, block phosphatidylserine on the surface of the EVs? In that case, the expression “Annexin V blocker” , for example, in the figures should be replaced by “Annexin V”. Please, clarify.

5- The title of section 2.5 is not clear and should be reformulated.

6- Section 2.5. The dimension of the pores of the transwells should be indicated as 0.4 μm and not 0.4 μ

7- Section 2.5. How many cells (neutrophils) were placed on the inserts?

8- Section 2.6 and section 2.10. Please, provide the catalogue number of all antibodies used as well as the catalogue number of the ELISA kit.

9- Section 2.12. MTT assay. The experimental setting is not clear. Can the MTT reagent also be reduced by the neutrophils present in the coculture? In that case, it could result in an interference. Please, clarify.

10- Section 3.1. The first sentence of the section is not clear. Please reformulate.

11- Section 3.1. Last line of the first paragraph. The reference should be to Supplementary Fig 1D (not 1C). Please, check.

12- Figure 1 (legend). Please, define the abbreviation fMLP.

13- Section 3.3. According to the text, the effect of MDA-EVs on Arginase 1 expression were not prevented by Annexin treatment. However, figure 3E shows a clear prevention of the increase in Arginase 1 expression in the group MDA-EV + Annexin V. Please, check.

14- Figure 3E. Beta-actin seems not to be a proper housekeeping protein. Specially, when a treatment is expected to affect the cell migration, as in this case, changes in the cytoskeleton can take place. Another normalization method should be used.

15- Further data supporting the effects on apoptosis is required (e.g., cleaved caspase 3 or cleaved PARP determination).

16- Supplementary information. “Sintenin-1” should be spelled “Syntenin-1”. Please, correct.

17- Section 3.4. The first sentence should be checked.

18- In general, the manuscript needs to be checked for grammar and syntax.

Author Response

We thank the reviewer for the opportunity to improve the manuscript. We have incorporated all suggestions in this new version.

Referee

1- Section 2.1. Specify that MCF10 cells are non-tumorigenic.

We thank you for your commentary. To fix this, we have inserted this specification on line 60.

Answer:

Section 2.1 / lines 60-61:

“Human non-tumor cell line MCF10 and human breast carcinoma cell line MDA-MB-231 were obtained from ATCC (Manassas, VA, USA).”

Referee:

2- Section 2.1. The method for EV isolation requires further explanation. How many cells were plated in order to obtained the CM? In which volume were the EVs resuspended? Please, specify which EV diluent was used for which application.

Answer:

Thanks for the observation. We had approximately 7x106 cells in each flask of 75 cm2 containing 10 mL of DMEM 1% FBS. After isolation, the EVs were resuspended in the original volume (10 mL). In general, EVs were resuspended in RPMI-1640 without FBS; for ROS assays, the EVs were resuspended HBSS; for western blotting, the EVs were resuspended in radioimmunoprecipitation lysis (RIPA) buffer, or for characterization by flow cytometry, the EVs were resuspended in a binding buffer to annexin-V

We have inserted this information in "Section 2.1 / lines 68-77", as can be seen below:

 “When the cells reached 100 % confluence (7x106 in bottle), the medium containing 10 % FBS was changed to 1 %, and the cells were kept for 24 h under the same conditions. Conditioned medium (CM) from tumor and non-tumor cells were collected after 24 h and centrifuged (1000 x g at 10 minutes) to remove cell debris. Afterward, the CM-containing EVs were ultracentrifuged at 4 °C for four hours at 100,000 g, and the supernatant was discarded. The vesicles were resuspended in original volume (10 mL) in Hank's Balanced Salt Solution (HBSS) for ROS assay, incomplete culture medium RPMI-1640 for other assays, a buffer of radioimmunoprecipitation lysis (RIPA) for western blotting of EVs, or binding buffer to annexin-V for characterization of EVs in flow cytometry. After isolation, the EVs were stored for up to 6 months at -80 °C.”

Referee:

3- Section 2.1. For the generation of CM, cells were cultured in the presence of 1% FBS. FBS is a common source of EVs. Was the FBS previously treated in order to eliminate this potential interference?

Answer:

Thanks for the comment. We have depleted EVs from FBS through ultracentrifugation applied EVs isolation methodology (100,000 x g, 4ºC, 4h). This information was added in the manuscript, as can be seen below:

Section 2.1 / lines 64-66

“The EVs contained in the serum were removed using the same EVs isolation protocol (described below). The EVs were discarded, and the supernatant was collected to be used in cell culture.”

Referee:

4- The use of the expression “Annexin-V blocker” throughout the manuscript is confusing. Was actually “Annexin V” used to bind and, therefore, block phosphatidylserine on the surface of the EVs? In that case, the expression “Annexin V blocker”, for example, in the figures should be replaced by “Annexin V”. Please, clarify.

Answer:

We thank the reviewer for this comment. We removed all mentions of the term blocker throughout the manuscript.

Referee:

5- The title of section 2.5 is not clear and should be reformulated.

Answer:

To address this issue, we have changed the title of the section to "Whole-cell extraction MDA-MB-231", as can be seen below:

Section 2.5 / line 110

“2.5 Whole-cell extraction MDA-MB-231”

Referee:

6- Section 2.5. The dimension of the pores of the transwells should be indicated as 0.4 μm and not 0.4 μ

Answer:

We thank the reviewer for calling our attention to this mistake. This issue was corrected in the manuscript:

Section 2.5 / line 115

“[…] and 0.4 µm transwell […]."

Referee:

7- Section 2.5. How many cells (neutrophils) were placed on the inserts?

Answer:

To address this question, we incorporated this information into the manuscript.

Section 2.5 / line 116

 “Neutrophils (106 cells/insert) were placed on top of the inserts […]."

Referee:

8- Section 2.6 and section 2.10. Please, provide the catalogue number of all antibodies used as well as the catalogue number of the ELISA kit.

Answer:

We thank the reviewer for bringing up this point. We added all information about the catalog numbers, as can be seen bellow:

Section 2.6 / lines 133-138

“[…] anti-arginase (gt5811 - rabbit - 1:1000) (Thermo Fisher Scientific), anti-caspase-3 (sc7148 - rabbit-1: 250) (Santa Cruz Biotechnology, Dallas, TX, USA), or anti-β-actin (a5441 - mouse - 1:5000) (Sigma-Aldrich). The membranes were incubated for at least one hour with a specific peroxidase-conjugated anti-rabbit secondary antibody (ab6789 - mouse or ab6721 […]."

Section 2.10 / lines 177 e 178

“[…] Development Kit from Peprotech (Rocky Hill, NJ, USA) (IL-8 #900-TM31; VEGF #900-TM10; CCL2 #900-TM31 and CCL5 #900-M33) […]”

Referee:

9- Section 2.12. MTT assay. The experimental setting is not clear. Can the MTT reagent also be reduced by the neutrophils present in the coculture? In that case, it could result in an interference. Please, clarify.

Answer:

To address this question, neutrophils were incubated alone with MTT, and the results have shown that they failed in metabolizing MTT. We included this information both in methodology and result section:

Section 2.13 / lines 217-218

“We also incubated only neutrophils with MTT to exclude the possibility of neutrophils metabolizing MTT.”

Section 3.4 / lines 341-342

“We also evaluated neutrophils alone; however, we observed that they could not metabolize MTT.”

Referee:

10- Section 3.1. The first sentence of the section is not clear. Please reformulate.

Answer:

We thank the reviewer for the opportunity to improve our manuscript. We have extensively reviewed the manuscript, which is now more straightforward and understandable.

Referee

11- Section 3.1. Last line of the first paragraph. The reference should be to Supplementary Fig 1D (not 1C). Please, check.

Answer:

Thanks for calling our attention to this point. We have fixed this issue.

Referee:

12- Figure 1 (legend). Please, define the abbreviation fMLP.

Answer:

Thanks for the observation. We added this information to the manuscript.

Figure legend 1, Line 280

 “fMLP: N-Formyl-L-methionyl-L-leucyl-L-phenylalanine”

Referee:

13- Section 3.3. According to the text, the effect of MDA-EVs on Arginase 1 expression were not prevented by Annexin treatment. However, figure 3E shows a clear prevention of the increase in Arginase 1 expression in the group MDA-EV + Annexin V. Please, check.

Answer:

We thank the reviewer for this observation. We reformulated this sentence.

Section 3.3 Line 306-309

“We also observed increased arginase-1 expression in neutrophils treated with MDA-EVs compared to those treated with MCF10-EVs, and this increase was not observed when MDA-EVs were pre-treated with annexin-V (Fig. 3E).”

Referee:

14- Figure 3E. Beta-actin seems not to be a proper housekeeping protein. Specially, when a treatment is expected to affect the cell migration, as in this case, changes in the cytoskeleton can take place. Another normalization method should be used.

Answer:

We completely agree with the reviewer. However, in previous works from our group and our lab routine, we observe that Beta-actin is a better housekeeping protein than others, such as alfa-tubulin. Below (in the left), we show the membrane stained with the ponceau rouge marker, apparently without any protein variation; in the immunoblotting image (in the right), we present the same membrane, labeled for α-tubulin (first bands in 55 kDa) and β-actin (in 42 kDa). As can be observed, the beta-actin appears to be a better housekeeping strategy with a better resolution.

<<<<Image>>

Referee

15- Further data supporting the effects on apoptosis is required (e.g., cleaved caspase 3 or cleaved PARP determination).

Answer

Thanks for the opportunity to improve our manuscript. We performed a new assay to investigate the mitochondrial membrane potential through the JC-1 probe. The JC-1 probe is widely used in apoptosis studies to monitor mitochondrial membrane integrity. Given the mitochondrial membrane potential changes during apoptosis, we observed that N2-like neutrophils increased MDA-MB-231 mitochondrial membrane potential integrity compared to neutrophils. The complete experimental methodology was described in the new section 2.14, lines 223-232:

2.14 Mitochondrial Membrane Potential Assay

“In a 24-well plate, 5x104 cells/well were plated in DMEM/F12 10% FBS. In the next step, the neutrophil purification was started, and the groups were prepared as previously described. After polarization, neutrophils were centrifuged at 1000 x g for 10 minutes, the supernatant discarded, and the cells resuspended in RPMI-1640 10% FBS. The culture medium was removed from all wells of the plate containing MDA-MB-231, and a 0.4 µm transwell was allocated to all wells. Neutrophils (5x105 cells/insert) were placed on top of the inserts. After 20 h of stimulation, the inserts with neutrophils were discarded as well as the medium remaining in the plate, and the tumor cells’ mitochondrial transmembrane potential was assessed in EnVision™ using JC-1 probe (10 μg/mL, Thermo Fisher Scientific), according to the manufacturer’s instructions.”

And the result was described in section 3.4, lines 344-348:

“We also investigated mitochondrial membrane potential through the JC-1 probe to confirm these findings. Thus, we observed that N2-like neutrophils inhibit MDA-MB-231 mitochondrial membrane potential enhancement induced by neutrophils. We also observed that neutrophils treated with LPS strongly disrupted MDA-MB-231 mitochondrial membrane potential (Fig. 4C).”

Referee

16- Supplementary information. “Sintenin-1” should be spelled “Syntenin-1”. Please, correct.

Answer

Thank you for the observation. We have corrected this issue.

Referee

17- Section 3.4. The first sentence should be checked.

Answer:

We thank the reviewer for bringing our attention to this point. In the new version of the manuscript, we corrected this sentence.

Section 3.4, lines 337-338

“We investigated the role of N2-like neutrophils upon breast tumor cell viability. For this, cells were treated with neutrophils previously incubated or not with EVs.”

Referee

18- In general, the manuscript needs to be checked for grammar and syntax.

Answer:

We thank the reviewer for the opportunity to improve our manuscript. We have performed an extensive review and grammar checking with the aid of Grammarly, a writing assistant tool.

Round 2

Reviewer 2 Report

The authors have addressed all my concerns. 

Author Response

(The authors gave the same response as above.)
